# Discriminative Gaifman Models

**Mathias Niepert**
NEC Labs Europe
Heidelberg, Germany
mathias.niepert@neclabs.eu

## Abstract

We present discriminative Gaifman models, a novel family of relational machine learning models. Gaifman models learn feature representations bottom up from representations of locally connected and bounded-size regions of knowledge bases (KBs). Considering local and bounded-size neighborhoods of knowledge bases renders logical inference and learning tractable, mitigates the problem of over-fitting, and facilitates weight sharing. Gaifman models sample neighborhoods of knowledge bases so as to make the learned relational models more robust to missing objects and relations which is a common situation in open-world KBs. We present the core ideas of Gaifman models and apply them to large-scale relational learning problems. We also discuss the ways in which Gaifman models relate to some existing relational machine learning approaches.

## 1 Introduction

Knowledge bases are attracting considerable interest both from industry and academia [2, 6, 15, 10]. Instances of knowledge bases are the web graph, social and citation networks, and multi-relational knowledge graphs such as Freebase [2] and YAGO [11]. Large knowledge bases motivate the development of scalable machine learning models that can reason about objects as well as their properties and relationships. Research in statistical relational learning (SRL) has focused on particular formalisms such as Markov logic [22] and PROBLOG [8] and is often concerned with improving the efficiency of inference and learning [14, 28]. The scalability problems of these statistical relational languages, however, remain an obstacle and have prevented a wider adoption. Another line of work focuses on efficient relational machine learning models that perform well on a particular task such as knowledge base completion and relation extraction. Examples are knowledge base factorization and embedding approaches [5, 21, 23, 26] and random-walk based ML models [15, 10]. We aim to advance the state of the art in relational machine learning by developing efficient models that learn knowledge base embeddings that are effective for probabilistic query answering on the one hand, and interpretable and widely applicable on the other.

Gaifman's locality theorem [9] is a result in the area of finite model theory [16]. The Gaifman graph of a knowledge base is the undirected graph whose nodes correspond to objects and in which two nodes are connected if the corresponding objects co-occur as arguments of some relation. Gaifman's locality theorem states that every first-order sentence is equivalent to a Boolean combination of sentences whose quantifiers range over local neighborhoods of the Gaifman graph. With this paper, we aim to explore Gaifman locality from a machine learning perspective. If every first-order sentence is equivalent to a Boolean combination of sentences whose quantifiers range over local neighborhoods only, we ought to be able to develop models that learn effective representations from these local neighborhoods. There is increasing evidence that learning representations that are built up from local structures can be highly successful. Convolutional neural networks, for instance, learn features over locally connected regions of images. The aim of this work is to investigate the effectiveness and efficiency of machine learning models that perform learning and inference within and across

locally connected regions of knowledge bases. This is achieved by combining relational features that are often used in statistical relatinal learning with novel ideas from the area of deep learning. The following problem motivates Gaifman models.

**Problem 1.** *Given a knowledge base (relational structure, mega-example, knowledge graph) or a collection of knowledge bases, learn a relational machine learning model that supports complex relational queries. The model learns a probability for each tuple in the query answer.*

Note that this is a more general problem than knowledge base completion since it includes the learning of a probability distribution for a complex relational query. The query corresponding to knowledge base completion is $\mathtt{r}(x, y)$ for logical variables $x$ and $y$, and relation $\mathtt{r}$. The problem also touches on the problem of open-world probabilistic KBs [7] since tuples whose prior probability is zero will often have a non-zero probability in the query answer.

## 2 Background

We first review some important concepts and notation in first-order logic.

### 2.1 Relational First-order Logic

An atom $\mathtt{r}(t_1, ..., t_n)$ consists of predicate $\mathtt{r}$ of arity $n$ followed by $n$ arguments, which are either elements from a finite domain $\mathbf{D} = \{a, b, ...\}$ or logical variables $\{x, y, ...\}$. We us the terms domain element and object synonymously. A ground atom is an atom without logical variables. Formulas are built from atoms using the usual Boolean connectives and existential and universal quantification. A free variable in a first-order formula is a variable $x$ not in the scope of a quantifier. We write $\varphi(x, y)$ to denote that $x, y$ are free in $\varphi$, and $\mathtt{free}(\varphi)$ to refer to the free variables of $\varphi$. A substitution replaces all occurrences of logical variable $x$ by $t$ in some formula $\varphi$ and is denoted by $\varphi[x/t]$.

A vocabulary consists of a finite set of predicates $\mathbf{R}$ and a domain $\mathbf{D}$. Every predicate $\mathtt{r}$ is associated with a positive integer called the arity of $\mathtt{r}$. A $\mathbf{R}$-structure (or knowledge base) $\mathcal{D}$ consists of the domain $\mathbf{D}$, a set of predicates $\mathbf{R}$, and an interpretation. The Herbrand base of $\mathcal{D}$ is the set of all ground atoms that can be constructed from $\mathbf{R}$ and $\mathbf{D}$. The interpretation assigns a truth value to every atom in the Herbrand base by specifying $\mathtt{r}^{\mathcal{D}} \subseteq \mathbf{D}^n$ for each $n$-ary predicate $\mathtt{r} \in \mathbf{R}$. For a formula $\varphi(x_1, ..., x_n)$ and a structure $\mathcal{D}$, we write $\mathcal{D} \models \varphi(d_1, ..., d_n)$ to say that $\mathcal{D}$ satisfies $\varphi$ if the variables $x_1, ..., x_n$ are substituted with the domain elements $d_1, ..., d_n$. We define $\varphi(\mathcal{D}) := \{(d_1, ..., d_n) \in \mathbf{D}^n \mid \mathcal{D} \models \varphi(d_1, ..., d_n)\}$. For the $\mathbf{R}$-structure $\mathcal{D}$ and $\mathbf{C} \subseteq \mathbf{D}$, $\langle \mathbf{C} \rangle^{\mathcal{D}}$ denotes the substructure induced by $\mathbf{C}$ on $\mathcal{D}$, that is, the $\mathbf{R}$-structure $\mathcal{C}$ with domain $\mathbf{C}$ and $\mathtt{r}^{\mathcal{C}} := \mathtt{r}^{\mathcal{D}} \cap \mathbf{C}^n$ for every $n$-ary $\mathtt{r} \in \mathbf{R}$.

### 2.2 Gaifman's Locality Theorem

The Gaifman graph of a $\mathcal{R}$-structure $\mathcal{D}$ is the graph $G_{\mathcal{D}}$ with vertex set $\mathbf{D}$ and an edge between two vertices $d, d' \in \mathbf{D}$ if and only if there exists an $\mathtt{r} \in \mathbf{R}$ and a tuple $(d_1, ..., d_k) \in \mathtt{r}^{\mathcal{D}}$ such that $d, d' \in \{d_1, ..., d_k\}$. Figure 1 depicts a fragment of a knowledge base and the corresponding Gaifman graph. The distance $\mathtt{d}_{\mathcal{D}}(d_1, d_2)$ between two elements $d_1, d_2 \in \mathbf{D}$ of a structure $\mathcal{D}$ is the length of the shortest path in $G_{\mathcal{D}}$ connecting $d_1$ and $d_2$. For $r \geq 1$ and $d \in \mathbf{D}$, we define the $r$-neighborhood of $d$ to be $\mathbf{N}_r(d) := \{x \in \mathbf{D} \mid \mathtt{d}_{\mathcal{D}}(d, x) \leq r\}$. We refer to $r$ also as the *depth* of the neighborhood. Let $\mathbf{d} = (d_1, ..., d_n) \in \mathbf{D}^n$. The $r$-neighborhood of $\mathbf{d}$ is defined as

$$\mathbf{N}_r(\mathbf{d}) = \bigcup_{i=1}^{n} \mathbf{N}_r(d_i).$$

For the Gaifman graph in Figure 1, we have that $\mathbf{N}_1(d_4) = \{d_1, d_2, d_5\}$ and $\mathbf{N}_1((d_1, d_2)) = \{d_1, ..., d_6\}$. $\varphi^{\mathbf{N}_r}(x)$ is the formula obtained from $\varphi(x)$ by *relativizing* all quantifiers to $\mathbf{N}_r(x)$, that is, by replacing every subformula of the form $\exists y \psi(x, y, \mathbf{z})$ by $\exists y (\mathtt{d}_{\mathcal{D}}(x, y) \leq r \wedge \psi(x, y, \mathbf{z}))$ and every subformula of the form $\forall y \psi(x, y, \mathbf{z})$ by $\forall y (\mathtt{d}_{\mathcal{D}}(x, y) \leq r \rightarrow \psi(x, y, \mathbf{z}))$. A formula $\psi(x)$ of the form $\varphi^{\mathbf{N}_r}(x)$, for some $\varphi(x)$, is called $r$-local. Whether an $r$-local formula $\psi(x)$ holds depends only on the $r$-neighborhood of $x$, that is, for every structure $\mathcal{D}$ and every $d \in \mathbf{D}$ we have $\mathcal{D} \models \psi(d)$

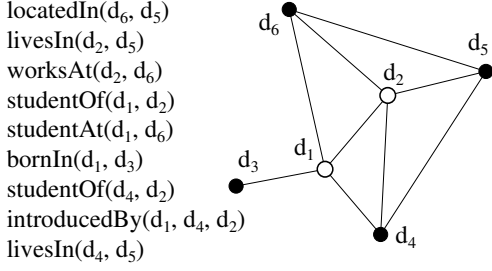

locatedIn($d_6, d_5$)
livesIn($d_2, d_5$)
worksAt($d_2, d_6$)
studentOf($d_1, d_2$)
studentAt($d_1, d_6$)
bornIn($d_1, d_3$)
studentOf($d_4, d_2$)
introducedBy($d_1, d_4, d_2$)
livesIn($d_4, d_5$)

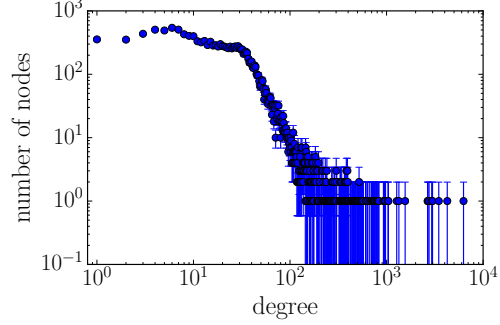

Figure 1: A knowledge base fragment for the pair $(d_1, d_2)$ and the corresponding Gaifman graph.

Figure 2: The degree distribution of the Gaifman graph for the Freebase fragment FB15K.

if and only if $\langle \mathbf{N}_r(d) \rangle \models \psi(d)$. For $r, k \geq 1$ and $\psi(x)$ being $r$-local, a local sentence is of the form

$$\exists x_1 \cdots \exists x_k \left( \bigwedge_{1 \leq i < j \leq k} \mathsf{d}_{\mathcal{D}}(x_i, x_j) > 2r \wedge \bigwedge_{1 \leq i \leq k} \psi(x_i) \right).$$

We can now state Gaifman's locality theorem.

**Theorem 1.** *[9] Every first-order sentence is equivalent to a Boolean combination of local sentences.*

Gaifman's locality theorem states that any first-order sentence can be expressed as a Boolean combination of $r$-local sentences defined for neighborhoods of objects that are mutually far apart (have distance at least $2r + 1$). Now, a novel approach to (statistical) relational learning would be to consider a large set of objects (or tuples of objects) and learn models from their local neighborhoods in the Gaifman graphs. It is this observation that motivates Gaifman models.

## 3  Learning Gaifman Models

Instead of taking the costly approach of applying relational learning and inference directly to entire knowledge bases, the representations of Gaifman models are learned bottom up, by performing inference and learning within bounded-size, locally connected regions of Gaifman graphs. Each Gaifman model specifies the data generating process from a given knowledge base (or collection of knowledge bases), a set of relational features, and a ML model class used for learning.

**Definition 1.** *Given a* $\mathbf{R}$*-structure* $\mathcal{D}$*, a discriminative Gaifman model for* $\mathcal{D}$ *is a tuple* $(\mathsf{q}, r, k, \mathbf{\Phi}, \mathcal{M})$ *as follows:*

- $\mathsf{q}$ *is a first-order formula called the* target query *with at least one free variable;*

- $r$ *is the* depth *of the Gaifman neighborhoods;*

- $k$ *is the* size-bound *of the Gaifman neighborhoods;*

- $\mathbf{\Phi}$ *is a set of first-order formulas (the relational features);*

- $\mathcal{M}$ *is the base model class (loss, hyper-parameters, etc.).*

Throughout the rest of the paper, we will provide detailed explanations of the different parameters of Gaifman models and their interaction with data generation, learning, and inference.

During the training of Gaifman models, neighborhoods are generated for *tuples of objects* $\mathbf{d} \in \mathbf{D}^n$ based on the parameters $r$ and $k$. We first describe the procedure for arbitrary tuples $\mathbf{d}$ of objects and will later explain where these tuples come from. For a given tuple $\mathbf{d}$ the $r$-neighborhood of $\mathbf{d}$ within the Gaifman graph is computed. This results in the set of objects $\mathbf{N}_r(\mathbf{d})$. Now, from this neighborhood we sample $w$ neighborhoods consisting of at most $k$ objects. Sampling bounded-size sub-neighborhoods from $\mathbf{N}_r(\mathbf{d})$ is motivated as follows:

1. The degree distribution of Gaifman graphs is often skewed (see Figure 2), that is, the number of other objects a domain element is related to varies heavily. Generating smaller, bounded-size neighborhoods allows the transfer of learned representations between more and less connected objects. Moreover, the sampling strategy makes Gaifman models more robust to object uncertainty [19]. We show empirically that larger values for $k$ reduce the effectiveness of the learned models for some knowledge bases.

2. Relational learning and inference is performed within the generated neighborhoods. $\mathbf{N}_r(\mathbf{d})$ can be very large, even for $r = 1$ (see Figure 2), and we want full control over the complexity of the computational problems.

3. Even for a single object tuple $\mathbf{d}$ we can generate a large number of training examples if $|\mathbf{N}_r(\mathbf{d})| > k$. This mitigates the risk of overfitting. The number of training examples per tuple strongly influences the models' accuracy.

We can now define the set of $(r, k)$-neighborhoods generated from a $r$-neighborhood.

$$\mathbf{N}_{r,k}(\mathbf{d}) := \begin{cases} \{\mathbf{N} \mid \mathbf{N} \subseteq \mathbf{N}_r(\mathbf{d}) \text{ and } |\mathbf{N}| = k\} & \text{if } |\mathbf{N}_r(\mathbf{d})| \geq k \\ \{\mathbf{N}_r(\mathbf{d})\} & \text{otherwise.} \end{cases}$$

For a given tuple of objects $\mathbf{d}$, Algorithm 1 returns a set of $w$ neighborhoods drawn from $\mathbf{N}_{r,k}(\mathbf{d})$ such that the number of objects for each $d_i$ is the same in expectation.

The formulas in the set $\mathbf{\Phi}$ are indexed and of the form $\varphi_i(s_1, ..., s_n, u_1, ..., u_m)$ with $s_j \in \mathtt{free}(\mathtt{q})$ and $u_j \notin \mathtt{free}(\mathtt{q})$. For every tuple $\mathbf{d} = (d_1, ..., d_n)$, generated neighborhood $\mathbf{N} \in \mathbf{N}_{r,k}(\mathbf{d})$, and $\varphi_i \in \mathbf{\Phi}$, we perform the substitution $[s_1/d_1, ..., s_n/d_n]$ and relativize $\varphi_i$'s quantifiers to $\mathbf{N}$, resulting in $\varphi_i^{\mathbf{N}}[s_1/d_1, ..., s_n/d_n]$ which we write as $\varphi_i^{\mathbf{N}}[\mathbf{s}/\mathbf{d}]$. Let $\langle \mathbf{N} \rangle$ be the substructure induced by $\mathbf{N}$ on $\mathcal{D}$. For every formula $\varphi_i(s_1, ..., s_n, u_1, ..., u_m)$ and every $\mathbf{n} \in \mathbf{N}^m$, we now have that $\mathcal{D} \models \varphi_i^{\mathbf{N}}[\mathbf{s}/\mathbf{d}, \mathbf{u}/\mathbf{n}]$ if and only if $\langle \mathbf{N} \rangle \models \varphi_i^{\mathbf{N}}[\mathbf{s}/\mathbf{d}, \mathbf{u}/\mathbf{n}]$. In other words, satisfaction is now checked locally within the neighborhoods $\mathbf{N}$, by deciding whether $\langle \mathbf{N} \rangle \models \varphi_i^{\mathbf{N}}[\mathbf{s}/\mathbf{d}, \mathbf{u}/\mathbf{n}]$. The relational semantics of Gaifman models is based on the set of formulas $\mathbf{\Phi}$. The feature vector $\mathbf{v} = (v_1, ..., v_{|\mathbf{\Phi}|})$ for tuple $\mathbf{d}$, and neighborhood $\mathbf{N} \in \mathbf{N}_{r,k}(\mathbf{d})$, written as $\mathbf{v_N}$, is constructed as follows

$$v_i := \begin{cases} \varphi_i^{\mathbf{N}}[\mathbf{s}/\mathbf{d}](\langle \mathbf{N} \rangle) & \text{if } \mathtt{free}(\varphi_i^{\mathbf{N}}[\mathbf{s}/\mathbf{d}]) > 0 \\ 1 & \text{if } \langle \mathbf{N} \rangle \models \varphi_i^{\mathbf{N}}[\mathbf{s}/\mathbf{d}] \\ 0 & \text{otherwise.} \end{cases}$$

That is, if $\varphi_i^{\mathbf{N}}[\mathbf{s}/\mathbf{d}]$ has free variables, $v_i$ is equal to the *number of groundings* of $\varphi_i[\mathbf{s}/\mathbf{d}]$ that are satisfied within the neighborhood substructure $\langle \mathbf{N} \rangle$; if $\varphi_i[\mathbf{s}/\mathbf{d}]$ has no free variables, $v_i = 1$ if and only if $\varphi_i[\mathbf{s}/\mathbf{d}]$ is satisfied within the neighborhod substructure $\langle \mathbf{N} \rangle$; and $v_i = 0$ otherwise. The neighborhood representations $\mathbf{v}$ capture $r$-local formulas and help the model learn formula combinations that are associated with negative and positive examples. For the right choices of the parameters $r$ and $k$, the neighborhood representations of Gaifman models capture the relational structure associated with positive and negative examples.

Deciding $\mathcal{D} \models \varphi$ for a structure $\mathcal{D}$ and a first-order formula $\varphi$ is referred to as *model checking* and computing $\varphi(\mathcal{D})$ is called $\varphi$-*counting*. The combined complexity of model checking is PSPACE-complete [29] and there exists a $||\mathcal{D}||^{O(||\varphi||)}$ algorithm for *both* problems where $|| \cdot ||$ is the size of an encoding. Clearly, for most real-world KBs this is not feasible. For Gaifman models, however, where the neighborhoods are bounded-size, typically $10 \leq |\mathbf{N}| = k \leq 100$, the above representation can be computed very efficiently for a large class of relational features. We can now state the following complexity result.

**Theorem 2.** *Let $\mathcal{D}$ be a relational structure (knowledge base), let $\overline{\mathtt{d}}$ be the size of the largest $r$-neighborhood of $\mathcal{D}$'s Gaifman graph, and let $\overline{\mathtt{s}}$ be the greatest encoding size of any formula in $\mathbf{\Phi}$. For a Gaifman model with parameters $r$ and $k$, the worst-case complexity for computing the feature representations of $N$ neighborhoods is $O(N(\overline{\mathtt{d}} + |\mathbf{\Phi}|k^{\overline{\mathtt{s}}}))$.*

Existing SRL approaches could be applied to the generated neighborhoods, treating each as a possible world for structure and parameter learning. However, our goal is to learn relational models that utilize embeddings computed by multi-layered neural networks.

**Algorithm 1** GENNEIGHS: Computes a list of $w$ neighborhoods of size $k$ for an input tuple $\mathbf{d}$.

1: **input:** tuple $\mathbf{d} \in \mathbf{D}^n$, parameters $r$, $k$, and $w$
2: $\mathbf{S} = [\,]$
3: **while** $|\mathbf{S}| < w$ **do**
4:     $S = \emptyset$
5:     $N = \mathbf{N}_r(\mathbf{d})$
6:     **for all** $i \in \{1, ..., n\}$ **do**
7:       $U = \min(\lfloor k/n \rfloor, |\mathbf{N}_r(d_i)|)$ elements sampled uniformly from $\mathbf{N}_r(d_i)$
8:       $N = N \setminus U$
9:       $S = S \cup U$
10:     $U = \min(|S| - k, |N|)$ elements sampled uniformly from $N$
11:     $S = S \cup U$
12:     $\mathbf{S} = \mathbf{S} + S$
13: **return** $\mathbf{S}$

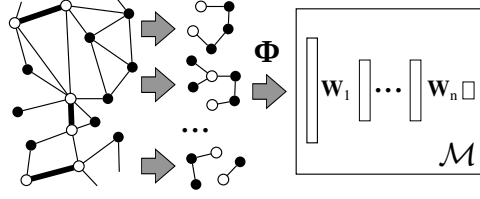

Figure 3: Learning of a Gaifman model.

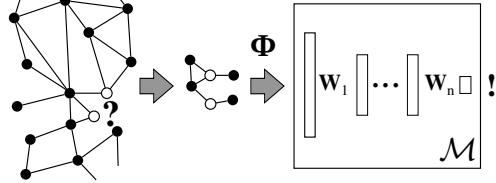

Figure 4: Inference with a Gaifman model.

### 3.1 Learning Distributions for Relational Queries

Let $\mathsf{q}$ be a first-order formula (the relational query) and $\mathcal{S}(\mathsf{q})$ the result set of the query, that is, all groundings that render the formula satisfied in the knowledge base. The feature representations generated for tuples of objects $\mathbf{d} \in \mathcal{S}(\mathsf{q})$ serve as *positive* training examples. The Gaifman models' aim is to learn neighborhood embeddings that capture local structure of tuples for which we know that the target query evaluates to true. Similar to previous work, we generate *negative* examples by corrupting tuples that correspond to positive examples. The corruption mechanism takes a positive input tuple $\mathbf{d} = (d_1, ..., d_n)$ and substitutes, for each $i \in \{1, ..., n\}$, the domain element $d_i$ with objects sampled from $\mathbf{D}$ while keeping the rest of the tuple fixed.

The discriminative Gaifman model performs the following steps.

1. Evaluate the target query $\mathsf{q}$ and compute the result set $\mathcal{S}(\mathsf{q})$

2. For each tuple $\mathbf{d}$ in the result set $\mathcal{S}(\mathsf{q})$:
   - Compute $\mathcal{N}$, a multiset of $w$ neighborhoods $\tilde{\mathbf{N}} \in \mathbf{N}_{r,k}(\mathbf{d})$ with Algorithm 1; each such neighborhood serves as a *positive* training example
   - Compute $\tilde{\mathcal{N}}$, a multiset of $\tilde{w}$ neighborhoods $\mathbf{N} \in \mathbf{N}_{r,k}(\tilde{\mathbf{d}})$ for corrupted versions of $\mathbf{d}$ with Algorithm 1; each such neighborhood serves as a *negative* training example
   - Perform model checking and counting within the neighborhoods to compute the feature representations $\mathbf{v_N}$ and $\mathbf{v_{\tilde{N}}}$ for each $\mathbf{N} \in \mathcal{N}$ and $\tilde{\mathbf{N}} \in \tilde{\mathcal{N}}$, respectively

3. Learn a ML model with the generated positive and negative training examples.

Learning the final Gaifman model depends on the base ML model class $\mathcal{M}$ and its loss function. We obtained state of the art results with neural networks, gradient-based learning, and categorical cross-entropy as loss function

$$\mathcal{L} = -\left[ \sum_{\mathbf{N} \in \mathcal{N}} \log p_{\mathcal{M}}(\mathbf{v_N}) + \sum_{\tilde{\mathbf{N}} \in \tilde{\mathcal{N}}} \log(1 - p_{\mathcal{M}}(\mathbf{v_{\tilde{N}}})) \right],$$

where $p_{\mathcal{M}}(\mathbf{v_N})$ is the probability the model returns on input $\mathbf{v_N}$. However, other loss functions are possible. The probability of a particular substitution of the target query to be true is now

$$P(\mathsf{q}[\mathbf{s}/\mathbf{d}] = \mathtt{True}) = \mathop{\mathbb{E}}_{\mathbf{N} \in \mathbf{N}_{(r,k)}(\mathbf{d})} [p_{\mathcal{M}}(\mathbf{v_N})].$$

The expected probability of a representation of a neighborhood drawn uniformly at random from $\mathbf{N}_{(r,k)}(\mathbf{d})$. It is now possible to generate several neighborhoods $\mathbf{N}$ and their representations $\mathbf{v_N}$ to

estimate $P(\mathbf{q}[\mathbf{s}/\mathbf{d}] = \texttt{True})$, simply by averaging the neighborhoods' probabilities. We have found experimentally that a single neighborhood already leads to highly accurate results but also that more neighborhood samples further improve the accurracy.

Let us emphasize again the novel semantics of Gaifman models. Gaifman models generate a large number of small, bounded-size structures from a large structure, learn a representation for these bounded-size structures, and use the resulting representation to answer queries concerning the original structure as a whole. The advantages are model weight sharing across a large number of neighborhoods and efficiency of the computational problems. Figure 3 and Figure 4 illustrate learning from bounded-size neighborhood structures and inference in Gaifman models.

## 3.2  Structure Learning

Structure learning is the problem of determining the set of relational features $\mathbf{\Phi}$. We provide some directions and leave the problem to future work. Given a collection of bounded-size neighborhoods of the Gaifman graph, the goal is to determine suitable relational features for the problem at hand. There is a set of features which we found to be highly effective. For example, formulas of the form $\exists x\ \mathbf{r}(s_1, x)$, $\exists x\ \mathbf{r}(s_1, x) \wedge \mathbf{r}(x, s_2)$, and $\exists x, y\ \mathbf{r}_1(s_1, x) \wedge \mathbf{r}_2(x, y) \wedge \mathbf{r}_3(y, s_2)$ for all relations. The latter formulas capture fixed-length paths between $s_1$ and $s_2$ in the neighborhoods. Hence, Path Ranking type features [15] can be used in Gaifman models as a particular relational feature class. For path formulas with several different relations we cannot include all $|\mathbf{R}|^3$ combinations and, hence, we have to determine a subset occurring in the training data. Fortunately, since the neighborhood size is bounded, it is computationally feasible to compute *frequent paths* in the neighborhoods and to use these as features. The complexity of this learning problem is in the number of elements in the neighborhood and not in the number of all objects in the knowledge base. Relation paths that do not occur in the data can be discarded. Gaifman models can also use features of the form $\forall x, y\ \mathbf{r}(x, y) \Rightarrow \mathbf{r}(y, x)$, $\exists x, y\ \mathbf{r}(x, y)$, and $\forall x, y, z\ \mathbf{r}(x, y) \wedge \mathbf{r}(y, z) \Rightarrow \mathbf{r}(x, z)$, to name but a few. Moreover, features with free variables, such as $\mathbf{r}(s_1, x)$ are *counting features* (here: the $\mathbf{r}$ out-degree of $s_1$). It is even computationally feasible to include specific second-order features (for instance, quantifiers ranging over $\mathbf{R}$) and aggregations of feature values.

## 3.3  Prior Confidence Values, Types, and Numerical Attributes

Numerous existing knowledge bases assign confidence values (probabilities, weights, etc.) to their statements. Gaifman models can incorporate confidence values during the sampling and learning process. Instead of adding random noise to the representations, which we have found to be beneficial, noise can be added inversely proportional to the confidence values. Statements for which the prior confidence values are lower are more likely to be dropped out during training than statements with higher confidence values. Furthermore, Gaifman models can directly incorporate object types such as *Actor* and *Action Movie* as well as numerical features such as *location* and *elevation*. One simply has to specify a fixed position in the neighborhood representation $\mathbf{v}$ for each object position within the input tuples $\mathbf{d}$.

## 4  Related Work

Recent work on relational machine learning for knowledge graphs is surveyed in [20]. We focus on a select few methods we deem most related to Gaifman models and refer the interested reader to the above article. A large body of work exists on learning inference rules from knowledge bases. Examples include [31] and [1] where inference rules of length one are learned; and [25] where general inference rules are learned by applying a support threshold. Their method does not scale to large KBs and depends on predetermined thresholds. Lao et al. [15] train a logistic regression classifier with path features to perform KB completion. The idea is to perform a random walk between objects and to exploit the discovered paths as features. SFE [10] improves PRA by making the generation of random walks more efficient. More recent embedding methods have combined paths in KBs with KB embedding methods [17]. Gaifman models support a much broader class of relational features subsuming path features. For instance, Gaifman models incorporate counting features that have shown to be beneficial for relational models.

Latent feature models learn features for objects and relations that are not directly observed in the data. Examples of latent feature models are tensor factorization [21, 23, 26] and embedding models [5, 3, 4, 18, 13, 27]. The majority of these models can be understood as more or less complex neural networks operating on object and relation representations. Gaifman models can also be used to learn knowledge base embeddings. Indeed, one can show that it generalizes or complements existing approaches. For instance, the universal schema [23] considers pairs of objects where relation membership variables comprise the model's features. We have the following interesting relationship between universal schemas [23] and Gaifman models. Given a knowledge base $\mathcal{D}$. The Gaifman model for $\mathcal{D}$ with $r = 0$, $k = 2$, $\Phi = \bigcup_{\mathbf{r} \in \mathbf{R}} \{\mathbf{r}(s_1, s_2), \mathbf{r}(s_2, s_1)\}$, $w = 1$ and $\tilde{w} = 0$ is equivalent to the Universal Schema [23] for $\mathcal{D}$ up to the base model class $\mathcal{M}$. More recent methods combine embedding methods and inference-based logical approaches for relation extraction [24]. Contrary to most existing multi-relational ML models [20], Gaifman models natively support higher-arity relations, functional and type constraints, numerical features, and complex target queries.

## 5  Experiments

The aim of the experiments is to understand the efficiency and effectiveness of Gaifman models for typical knowledge base inference problems. We evaluate the proposed class of models with two data sets derived from the knowledge bases WORDNET

Table 1: The statistics of the data sets.

| Dataset | $|\mathbf{D}|$ | $|\mathbf{R}|$ | # train | # test |
|---|---|---|---|---|
| WN18 | 40,943 | 18 | 141,442 | 5,000 |
| FB15k | 14,951 | 1,345 | 483,142 | 59,071 |

and FREEBASE [2]. Both data sets consist of a list of statements $\mathbf{r}(d_1, d_2)$ that are known to be true. For a detailed description of the data sets, whose statistics are listed in Table 1, we refer the reader to previous work [4].

After training the models, we perform entity prediction as follows. For each statement $\mathbf{r}(d_1, d_2)$ in the test set, $d_2$ is replaced by each of the KB's objects in turn. The probabilities of the resulting statements are predicted and sorted in descending order. Finally, the rank of the correct statement within this ordered list is determined. The same process is repeated now with replacements of $d_1$. We compare Gaifman models with $\mathbf{q} = \mathbf{r}(x, y)$ to state of the art knowledge base completion approaches which are listed in Table 2. We trained Gaifman models with $r = 1$ and different values for $k$, $w$, and $\tilde{w}$. We use a neural network architecture with two hidden layers, each having 100 units and sigmoid activations, dropout of 0.2 on the input layer, and a softmax layer. Dropout makes the model more robust to missing relations between objects. We trained one model per relation and left the hyper-parameters fixed across models. We did not perform structure learning and instead used the following set of relational features

$$\Phi := \bigcup_{\mathbf{r} \in \mathbf{R}, \, i \in \{1,2\}} \left\{ \begin{array}{l} \mathbf{r}(s_1, s_2), \mathbf{r}(s_2, s_1), \exists x \, \mathbf{r}(x, s_i), \exists x \, \mathbf{r}(s_i, x), \\ \exists x \, \mathbf{r}(s_1, x) \wedge \mathbf{r}(x, s_2), \exists x \, \mathbf{r}(s_2, x) \wedge \mathbf{r}(x, s_1) \end{array} \right\}.$$

To compute the probabilities, we averaged the probabilities of $N = 1, 2$, or $3$ generated $(r, k)$-neighborhoods.

We performed runtime experiments to evaluate the models' efficiency. Embedding models have the advantage that one dot product for every candidate object is sufficient to compute the score for the corresponding statement and we need to assess the performance of Gaifman models in this context. All experiments were run on commodity hardware with 64G RAM and a single 2.8 GHz CPU.

Table 2 lists the experimental results for different parameter settings $[N, k, w, \tilde{w}]$. The Gaifman models achieve the highest hits@10 and hits@1 values for both data sets. As expected, the more neighborhood samples are used to compute the probability estimate ($N = 1, 2, 3$) the better the result. When the entire 1-neighborhood is considered ($k = \infty$),

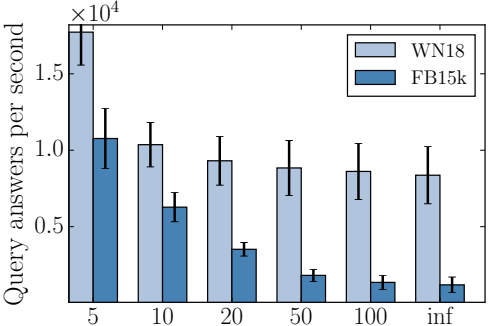

Figure 5: Query answers per second rates for different values of the parameter $k$.

the performance for WN18 does not deteriorate as it does for FB15k. This is due to the fact that

Table 2: Results of the entity prediction experiments.

| Data Set | WN18 | | | FB15K | | |
|---|---|---|---|---|---|---|
| Metric | Mean rank | Hits@10 | Hits@1 | Mean rank | Hits@10 | Hits@1 |
| RESCAL[21] | 1,163 | 52.8 | - | 683 | 44.1 | - |
| SE[5] | 985 | 80.5 | - | 162 | 39.8 | - |
| LFM[12] | 456 | 81.6 | - | 164 | 33.1 | - |
| TransE[4] | 251 | 89.2 | 8.9 | 51 | 71.5 | 28.1 |
| TransR[18] | 219 | 91.7 | - | 78 | 65.5 | - |
| DistMult[30] | 902 | 93.7 | 76.1 | 97 | 82.8 | 44.3 |
| Gaifman [1, $\infty$, 1, 5] | 298 | 93.9 | 75.8 | 124 | 78.1 | 59.8 |
| Gaifman [1, 20, 1, 2] | 357 | 88.1 | 66.8 | 114 | 79.2 | 60.1 |
| Gaifman [1, 20, 5, 25] | 392 | 93.6 | 76.4 | 97 | 82.1 | 65.6 |
| Gaifman [2, 20, 5, 25] | 378 | 93.9 | 76.7 | 84 | 83.4 | 68.5 |
| Gaifman [3, 20, 5, 25] | 352 | 93.9 | 76.1 | 75 | 84.2 | 69.2 |

objects in WN18 have on average few neighbors. FB15k has more variance in the Gaifman graph's degree distribution (see Figure 2) which is reflected in the better performance for smaller $k$ values. The experiments also show that it is beneficial to generate a large number of representations (both positive and negative ones). The performance improves with larger number of training examples.

The runtime experiments demonstrate that Gaifman models perform inference very efficiently for $k \leq 20$. Figure 5 depicts the number of query answers the Gaifman models are able to serve per second, averaged over relation types. A query answer returns the probability for one object pair. These numbers include neighborhood generation and network inference. The results are promising with about 5000 query answers per second (averaged across relation types) as long as $k$ remains small. Since most object pairs of WN18 have a 1-neighborhood whose size is smaller than 20, the answers per second rates for $k > 20$ is not reduced as drastically as for FB15k.

## 6   Conclusion and Future Work

Gaifman models are a novel family of relational machine learning models that perform learning and inference within and across locally connected regions of relational structures. Future directions of research include structure learning, more sophisticated base model classes, and application of Gaifman models to additional relational ML problems.

### Acknowledgements

Many thanks to Alberto García-Durán, Mohamed Ahmed, and Kristian Kersting for their helpful feedback.

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
