[Reviews · NeurIPS 2016]

Reviewer 1

Summary

This paper addresses the problem of learning relations through Gaifman models. The basic idea is that by converting a set of relations to a graph between atomic relations one can then turn the problem of learning predicates to

Qualitative Assessment

This paper is quite difficult to understand for anyone outside of logic. Ideas like relativizing are new to me, and makes the paper very difficult to follow. Given the space constraints this seems unavoidable. The experiments are convincing, and seem comprehensive.

Confidence in this Review

1-Less confident (might not have understood significant parts)


Reviewer 2

Summary

This paper applies discriminative Gaifman models to medium-sized public datasets for entity prediction.

Qualitative Assessment

I have to apologize for not understanding everything in the paper. Overall, the results show that on the datasets tested the Gaifman models perform much better than the tested competing approaches. However, the datasets do not seem to be sufficiently large to warrant general applicability in industrial settings as is. Since the models (and possibly the problems they can solve) are not mainstream it may be good to find clear applications to prove their usefulness for real problems.

Confidence in this Review

1-Less confident (might not have understood significant parts)


Reviewer 3

Summary

This paper proposes Gaifman models for relational learning. The core idea is to learn representations for knowledge graphs in a bottom up fashion starting from locally connected regions. Experimental results on two real-world datasets are presented to demonstrate effectiveness of the proposed model.

Qualitative Assessment

The paper was hard to read and it is clearly not ready as-is. While the core ideas of the paper are simple, I felt many sections were unnecessarily dense and could have been explained using much simpler representation. Also, this is not the first paper which has looked at local structure for learning and inference over knowledge graphs. For example, the recent SFE paper [1] does something similar, at least in spirit. Unfortunately, the paper doesn't connect to such important recent work which make it hard to evaluate true significance of the proposed work. Frankly I feel proposing a whole new formalism to achieve the core objectives in the paper is an overkill. It is also not clear what representation is being learnt in the query-driven model. [1] https://www.semanticscholar.org/paper/Efficient-and-Expressive-Knowledge-Base-Completion-Gardner-Mitchell/5b9c2b3f85920bc0e160b484ffa7a5f0a9d8f22a

Confidence in this Review

1-Less confident (might not have understood significant parts)


Reviewer 4

Summary

The paper present Gaifman models that combined Gaifman’s locality theorem with relational machine learning models. It perform learning and inference within and across locally connected regions of relational structures.

Qualitative Assessment

The research problem of this paper is interesting and meaningful. The paper present a novel family of relational machine learning models. However, it would be better if some flaws of the paper can be modified. 1. In line 20, the paper write "We aim to advance the .. models which learn efficient and effective cs". What are the shortcomings of existing methods in terms of efficiency and representation? 2.In line 20, the paper write "...learn effective representations from these local neighborhoods". What is " effective representations"? Could you give an example? 3.In line 143, the paper give a theorem 2. Could you give the proof of the theorem? 4. In line 20, the paper write " We obtained ... categorical cross-entropy as loss function ".Why use it? Could you give a reasonable analysis or advantage analysis? 5. There are similarities with PTranE(Modeling Relation Paths for Representation Learning of Knowledge Bases). Could you compare with this method? In addition, TransE and TransR are typical Knowledge Graph embedding model. But ,at present, the best model of Knowledge Graph Embedding is TranSparse (Knowledge Graph Completion with Adaptive Sparse Transfer Matrix). Could you compare with this method in the experiment to illustrate effective representations?

Confidence in this Review

2-Confident (read it all; understood it all reasonably well)


Reviewer 5

Summary

Using Gaifman's locality theorem, the paper propose a model to learn and infer locally for computational efficiency.

Qualitative Assessment

1. The paper is difficult to read due to its writing style. While it is formally written, it lacks high level description for more general reader. 2. I have a hard time finding what is a discriminative Gaifman models. 3. It looks like a paper with contribution, but might not clearly stated.

Confidence in this Review

1-Less confident (might not have understood significant parts)


Reviewer 6

Summary

The authors proposed a new approach for statistical relational learning over knowledge bases, where the feature of input tuples are augmented by their local neighbors on the Gaifman manifold during the training process. Encouraging results were obtained on Freebase and WordNet as compared to several representative baselines.

Qualitative Assessment

The way how feature vectors are constructed based on the local region of Gaifman manifold seems both reasonable and novel (as far as I'm aware), backed up by promising empirical performance. Both the motivation and the algorithmic solutions are presented in a technically sound manner. Overall I think this is a nice contribution. Questions: 1. The empirical evaluation is conducted under a special case of the proposed framework where each relation involves only a pair of arguments (probably in order to make the results comparable to existing baselines). I'm curious how the algorithm would actually perform under more generic scenarios (where each relation has more than two arguments), and whether this will bring about scalability issues. 2. To my understanding, the Gaifman manifold could be a weighted graph as clearly different entities are not equally correlated. Is there a particular reason to ignore the edge strength? 3. Line 71: N_r --> N_1?

Confidence in this Review

2-Confident (read it all; understood it all reasonably well)